# CANARY IN A COALMINE: BETTER MEMBERSHIP IN-FERENCE WITH ENSEMBLED ADVERSARIAL QUERIES

**Yuxin Wen**
University of Maryland
ywen@umd.edu

**Arpit Bansal**
University of Maryland

**Hamid Kazemi**
University of Maryland

**Eitan Borgnia**
University of Chicago

**Micah Goldblum**
New York University

**Jonas Geiping**
University of Maryland

**Tom Goldstein**
University of Maryland

## ABSTRACT

As industrial applications are increasingly automated by machine learning models, enforcing personal data ownership and intellectual property rights requires tracing training data back to their rightful owners. *Membership inference* algorithms approach this problem by using statistical techniques to discern whether a target sample was included in a model's training set. However, existing methods *only* utilize the unaltered target sample or simple augmentations of the target to compute statistics. Such a sparse sampling of the model's behavior carries little information, leading to poor inference capabilities. In this work, we use adversarial tools to directly optimize for queries that are discriminative and diverse. Our improvements achieve significantly more accurate membership inference than existing methods, especially in offline scenarios and in the low false-positive regime which is critical in legal settings. Code is available at https://github.com/YuxinWenRick/canary-in-a-coalmine

## 1 INTRODUCTION

In an increasingly data-driven world, legislators have begun developing a slew of regulations with the intention of protecting data ownership. The right-to-be-forgotten written into the strict GDPR law passed by the European Union has important implications for the operation of ML-as-a-service (MLaaS) providers (Wilka et al., 2017; Truong et al., 2021). As one example, Veale et al. (2018) discuss that machine learning models could legally (in terms of the GDPR) fall into the category of "personal data", which equips all parties represented in the data with rights to restrict processing and to object to their inclusion. However, such rights are vacuous if enforcement agencies are unable to detect when they are violated. Membership inference algorithms are designed to determine whether a given data point was present in the training data of a model. Though membership inference is often presented as a breach of privacy in situations where belonging to a dataset is itself sensitive information (*e.g.* a model trained on a group of people with a rare disease), such methods can also be used as a legal tool against a non-compliant or malicious MLaaS provider.

Because membership inference is a difficult task, the typical setting for existing work is generous to the attacker and assumes full white-box access to model weights. In the aforementioned legal scenario, this is not a realistic assumption. Organizations have an understandable interest in keeping their proprietary model weights secret and short of a legal search warrant, often only provide black-box querying to their clients (OpenAI, 2020). Moreover, even if a regulatory agency forcibly obtained white-box access via an audit, for example, a malicious provider could adversarially spoof the reported weights to cover up any violations.

In this paper, we achieve state-of-the-art performance for membership inference in the black-box setting by using a new adversarial approach. We observe that previous work (Shokri et al., 2017;

Yeom et al., 2018; Salem et al., 2018; Carlini et al., 2022a) improves membership inference attacks through a variety of creative strategies, but these methods query the targeted model using only the original target data point or its augmentations. We instead *learn* query vectors that are maximally discriminative; they separate all models trained with the target data point from all models trained without it. We show that this strategy reliably results in more precise predictions than the baseline method for three different datasets, four different model architectures, and even models trained with differential privacy.

## 2 BACKGROUND AND RELATED WORK

Homer et al. (2008) originated the idea of membership inference attacks (MIAs) by using aggregated information about SNPs to isolate a specific genome present in the underlying dataset with high probability. Such attacks on genomics data are facilitated by small sample sizes and the richness of information present in each DNA sequence, which for humans can be up to three billion base pairs. Similarly, the overparametrized regime of deep learning makes it vulnerable to MIAs. Yeom et al. (2018) designed the first attacks on deep neural networks by leveraging overfitting to the training data – *members* exhibit statistically lower loss values than *non-members*.

Since their inception, improved MIAs have been developed, across different problem settings and threat models with varying levels of adversarial knowledge. Broadly speaking, MIAs can be categorized into *metric-based* approaches and *binary classifier* approaches (Hu et al., 2021). The latter utilizes a variety of calculated statistics to ascertain membership while the former involves training shadow models and using a neural network to learn the correlation (Shokri et al., 2017; Truong et al., 2021; Salem et al., 2018).

More specifically, existing metric-based approaches include: correctness (Yeom et al., 2018; Choquette-Choo et al., 2021; Bentley et al., 2020; Irolla & Châtel, 2019; Sablayrolles et al., 2019), loss (Yeom et al., 2018; Sablayrolles et al., 2019), confidence (Salem et al., 2018), and entropy (Song & Mittal, 2021; Salem et al., 2018). The ability to query such metrics at various points during training has been shown to further improve membership inference. Liu et al. (2022) devise a model distillation approach to simulate the loss trajectories during training, and Jagielski et al. (2022b) leverage continual updates to model parameters to get multiple trajectory points.

Despite the vast literature on MIAs, all existing methods in both categories rely solely on the data point $x$ whose membership status is in question – metric-based approaches compute statistics based on $x^*$ or augmentations of $x^*$ and binary classifiers take $x^*$ as an input and output membership status directly. Our work hinges on the observation that an optimized *canary* image $x_{mal}$ can be a more effective litmus test for determining the membership of $x^*$. Note that this terminology is separate from the use in Zanella-Béguelin et al. (2020) and Carlini et al. (2019), where a canary refers to a sequence that serves as a proxy to test memorization of sensitive data in language models. It also differs from the canary-based gradient attack in Pasquini et al. (2021), where a malicious federated learning server sends adversarial weights to users to infer properties about individual user data (*e.g.* membership inference) even with secure aggregation.

The metric used for assessing the efficacy of a MIA has been the subject of some debate. A commonly used approach is *balanced attack accuracy*, which is an empirically determined probability of correctly ascertaining membership. However, Carlini et al. (2022a) point out that this metric is inadequate because it implicitly assigns equal weight to both classes of mistakes (*i.e.* false positive and false negatives) and it is an average-case metric. The latter characteristic is especially troubling because meaningful privacy should protect minorities and not be measured solely on effectiveness for the majority. A good alternative to address these shortcomings is to provide the *receiver operating characteristic* (ROC) curve. This metric reports the true positive rate (TPR) at each false positive rate (FPR) by varying the detection threshold. One way to distill the information present in the ROC curve is by computing the area under the curve (AUC) – more area means a higher TPR across all FPRs on average. However, more meaningful violations of privacy occur at a low FPR. Methods that optimize solely for AUC can overstress the importance of high TPR at high FPR, a regime inherently protected by plausible deniability. In our work, we report both AUC and numerical results at the FPR deemed acceptable by Carlini et al. (2022a) for ease of comparison.

There have been efforts to characterize the types of models and data most vulnerable to the MIAs described above. Empirical work has shown the increased privacy risk for more overparametrized models (Yeom et al., 2018; Carlini et al., 2022a; Leino & Fredrikson, 2020), which was made rigorous by Tan et al. (2022b) for linear regression models with Gaussian data. Tan et al. (2022a) show the overparameterization/privacy tradeoff can be improved by using wider networks and early stopping to prevent overfitting. From the data point of view, Carlini et al. (2022c) show that data at the tails of the training distribution are more vulnerable, and efforts to side-step the privacy leakage by removing tail data points just creates a new set of vulnerable data. Jagielski et al. (2022a) show data points encountered early in training are "forgotten" and thus more protected from MIAs than data encountered late in training.

## 3 LETTING THE CANARY FLY

In this section, we expound upon the threat model for the type of membership inference we perform. We then provide additional background on metric-based MIA through likelihood ratio tests, before describing how to optimize the `canary` query data point.

### 3.1 THREAT MODELS

Membership inference is a useful tool in many real-world scenarios. For example, suppose a MLaaS company trains an image classifier by scraping large amounts of online images and using data from users/clients to maximize model performance. A client requests that their data be unlearned from the company's model – via their right-to-be-forgotten – and wants to test compliance by determining membership inference of a private image during training. We assume the client also has the ability to scrape online data points, which may or may not be in the training data of the target classifier. However, the target model can only be accessed through an API that returns predictions and confidence scores, hiding weights and intermediate activations.

We formulate two threat models, where the *trainer* is the company and the *attacker* is the client as described above:

**Online Threat Model.** We assume there exists a public training algorithm $\mathcal{T}$ (including the model architecture) and a universal dataset $D$. The trainer trains a target model $\theta_\text{t}$ on a random subset $D_\text{t} \subseteq D$ through $\mathcal{T}$. Given a sensitive point $(x^*, y^*) \in D$, the attacker aims to determine whether $(x^*, y^*) \in D_\text{t}$ or $(x^*, y^*) \notin D_\text{t}$. The target model parameters are protected, and the attacker has limited query access to the target model and its confidence $f_{\theta_\text{t}}(x)_y$ for any $(x, y)$.

We use the term *online* to indicate that the attacker can modify their membership inference strategy as a function of $(x^*, y^*)$. A more conservative threat model is the *offline* variant, where the attacker must *a priori* decide on a fixed strategy to utilize across all sensitive data points. This is more realistic when the strategy involves training many shadow models, which is computationally expensive.

**Offline Threat Model.** As above, the trainer trains a target model on $D_\text{t} \subseteq D$ with $\mathcal{T}$. However, now we assume the attacker only has access to an auxiliary dataset $D_\text{aux} \subseteq D$ to prepare their attack. The set of sensitive data points $D_\text{test} \subseteq D$ is defined to have the properties $D_\text{aux} \cap D_\text{test} = \emptyset$ but $D_\text{t} \cap D_\text{test} \neq \emptyset$. Again, the attacker has limited query access to the target model and its confidence $f_{\theta_\text{t}}(x)_y$ for any $(x, y)$.

### 3.2 LIKELIHOOD RATIO ATTACKS

As a baseline, we start out with the metric-based Likelihood Ratio Attack (LiRA) introduced by Carlini et al. (2022a). In the **online** threat model, a LiRA attacker first trains $N$ shadow models $S = \{\theta_1, ..., \theta_N\}$ on randomized even splits of the dataset $D$. For any data point $(x, y) \in D$, it follows that there are on average $N/2$ OUT shadow models trained *without* $(x, y)$ and $N/2$ IN shadow models trained *with* $(x, y)$. This allows the attacker to run membership inference using a joint pool of shadow models, without having to retrain models for every new trial data point. Given a target point $x^*$ and its label $y^*$, an attacker calculates confidence scores of IN models

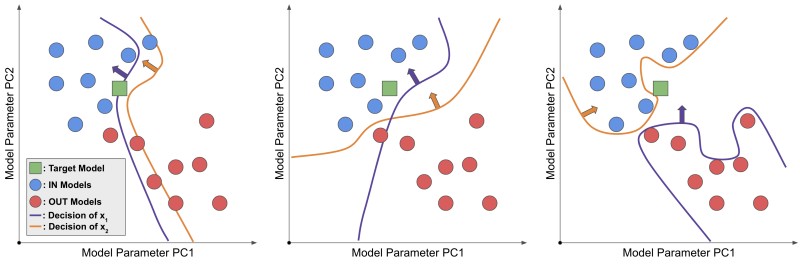

(a) Augmented Queries     (b) Adversarial Queries     (c) Overfitted Adv. Queries

Figure 1: **Query Decision Boundary in Model Parameter Space.** We illustrate our idea by plotting the decision boundaries of two queries $x_1$ and $x_2$ in *model parameter* space. In this case, the target image is indeed in the training data of (green) target model. We sketch three cases. (a) Both augmented queries are unable to separate both distributions and the membership inference fails. (b) Two optimal adversarial queries are generated that are both diverse and, on average, separate both distributions and the attack succeeds. (c) Without the constraints, adversarial queries can overfit and again lead to attack failure.

$S_{\text{in}} = \{\theta_1^{\text{in}}, ..., \theta_n^{\text{in}}\}$ and OUT models $S_{\text{out}} = \{\theta_1^{\text{out}}, ..., \theta_m^{\text{out}}\}$. Confidence scores are scaled via

$$\phi(f_\theta(x^*)_{y^*}) = \log(\frac{f_\theta(x^*)_{y^*}}{1 - f_\theta(x^*)_{y^*}}), \tag{1}$$

where $f_\theta(x)_y$ denotes the confidence score from the model $\theta$ on the point $(x, y)$. This scaling approximately standardizes the confidence distribution, as the distribution of the unnormalized confidence scores is often non-Gaussian. After retrieving the scaled scores for IN and OUT models, the attacker fits them to two separate Gaussian distributions denoted $\mathcal{N}(\mu_{\text{in}}, \sigma_{\text{in}}^2)$ and $\mathcal{N}(\mu_{\text{out}}, \sigma_{\text{out}}^2)$ respectively. Then, the attacker queries the target model with $(x^*, y^*)$ and computes the scaled confidence score of the target model $\text{conf}_{\text{t}} = \phi(f_{\theta_{\text{t}}}(x^*)_{y^*})$. Finally, the probability of $(x^*, y^*)$ being in the training data of $\theta_{\text{t}}$ is calculated as:

$$\frac{p(\text{conf}_{\text{t}} \mid \mathcal{N}(\mu_{\text{in}}, \sigma_{\text{in}}^2))}{p(\text{conf}_{\text{t}} \mid \mathcal{N}(\mu_{\text{out}}, \sigma_{\text{out}}^2))}, \tag{2}$$

where $p(\text{conf} \mid \mathcal{N}(\mu, \sigma^2))$ calculates the probability of $\text{conf}$ under $\mathcal{N}(\mu, \sigma^2)$.

For the **offline** threat model, the attacker exclusively produces OUT shadow models by training on a set of randomized datasets fully disjoint from the possible sensitive data. For the sensitive data point $(x^*, y^*)$, the final score is now calculated as a one-sided hypothesis which yields:

$$1 - p(\text{conf}_{\text{t}} \mid \mathcal{N}(\mu_{\text{out}}, \sigma_{\text{out}}^2))$$

Though assessing membership this way is more challenging, the offline model allows the attacker to avoid having to train any new models at inference time in response to a new $(x^*, y^*)$ pair – a more realistic setting if the attacker is a regulatory agency responding to malpractice claims by many users, for example.

In practice, modern machine learning models are trained with data augmentations. Both the online and offline methods above can be improved if the attacker generates $k$ augmented target data points $\{x_1, ..., x_k\}$, performs the above probability test on each of the $k$ augmented samples, and averages the resulting scores.

### 3.3 MOTIVATION

Despite achieving state-of-the-art results, a LiRA attacker *exclusively* queries the target model with the target data $(x^*, y^*)$ or simple, predefined augmentations of it. Even in the online setting, if $(x^*, y^*)$ is not an outlier that strongly influences the final IN model (Carlini et al., 2022c; Ilyas et al., 2022), then its impact on the model and thus the information gained from its confidence score is quite limited. Moreover, the longer a model trains, the further it becomes invariant to its data augmentations, so the ensemble of augmented target samples might still lack sufficient information to ascertain membership.

Since the threat model does not bar the attacker from querying arbitrary data, we ask whether more information about the target model can be obtained by *synthesizing* more powerful queries. Intuitively, we want the final synthesized query to always give statistically separable signals for models trained with/without the original sensitive sample. Existing work has shown two models trained with the same data point, often share similar properties at the decision boundary near that point (Somepalli et al., 2022). Using the shadow models from LiRA, the attacker can therefore adversarially optimize a query (near the original $x^*$) such that the distribution of confidence scores for IN models is as different as possible from the distribution of confidence scores for OUT models. We call the synthesized query a *canary* because it is an indicator for the membership status of $x^*$.

### 3.4 OPTIMIZING FOR CANARY SUCCESS

We now formally present our strategy to generate adversarial queries. For a target data point $(x^*, y^*)$, its IN shadow models $S_{\text{in}} = \{\theta_1^{\text{in}}, ..., \theta_n^{\text{in}}\}$, and its OUT shadow models $S_{\text{out}} = \{\theta_1^{\text{out}}, ..., \theta_m^{\text{out}}\}$, the attacker's goal is to find a data point $x_{\text{mal}}$ such that IN models and OUT models have different behaviors (logits/confidence scores/losses). In the simplest case, the attacker can optimize $x_{\text{mal}}$ so that IN shadow models have high losses on $x_{\text{mal}}$ and OUT models to have low losses on $x_{\text{mal}}$. This can be simply achieved by minimizing the following objective:

$$\underset{x_{\text{mal}} \in I}{\text{argmin}} \frac{1}{n} \sum_{i=1}^{n} \mathcal{L}(x_{\text{mal}}, y^*, \theta_i^{\text{in}}) + \frac{1}{m} \sum_{i=1}^{m} \mathcal{L}_{\text{out}}(x_{\text{mal}}, y^*, \theta_i^{\text{out}}), \tag{3}$$

where $I$ is the feasible data point domain, $\mathcal{L}$ is the main task loss, and $\mathcal{L}_{\text{out}}$ is $-\log(1 - f_\theta(x)_y)$. We further evaluate the optimal choice of objective functions in Section 4.4.

Though in principle an attacker can construct a canary query as described above, in practice the optimization problem is intractable. Accumulating the loss on all shadow models requires a significant amount of computational resources, especially for a large number of shadow models or models with many parameters. Another way to conceptualize the problem at hand, is to think of $x_{\text{mal}}$ as the model parameters and the shadow models as training data points in traditional machine learning. When framed this way, the number of parameters in our model $x_{\text{mal}}$ is much greater than the number of data points $|S_{\text{in}}| + |S_{\text{out}}|$.

---

**Algorithm 1** `Canary Algorithm`

---

1: **Input:** IN shadow models $S_{\text{in}} = \{\theta_1^{\text{in}}, ..., \theta_n^{\text{in}}\}$, OUT shadow models $S_{\text{out}} = \{\theta_1^{\text{out}}, ..., \theta_m^{\text{out}}\}$, target data point $(x^*, y^*)$, batch size $b$, optimization steps $T$, perturbation bound $\varepsilon$, input domain $I$
2: $\Delta_{\text{mal}} = 0$
3: **for** $1, ..., T$ **do**
4:     Shuffle the index for $S_{\text{in}}$ and $S_{\text{out}}$
5:     Calculate loss on OUT models:
6:     $g_{\Delta_{mal}} = \nabla_{\Delta_{mal}} \left[ \frac{1}{b} \sum_{i=1}^{b} \mathcal{L}_{\text{out}}(x^* + \Delta_{\text{mal}}, y^*, \theta_i^{\text{out}}) \right]$
7:     Calculate loss on IN models (removed when offline):
8:     $g_{\Delta_{mal}} \mathrel{+}= \nabla_{\Delta_{mal}} \left[ \frac{1}{b} \sum_{i=1}^{b} \mathcal{L}(x^* + \Delta_{\text{mal}}, y^*, \theta_i^{\text{in}}) \right]$
9:     Update $\Delta_{\text{mal}}$ based on $g_{\Delta_{\text{mal}}}$
10:    Project $\Delta_{\text{mal}}$ onto $||\Delta_{\text{mal}}||_\infty \leq \varepsilon$ and $(x^* + \Delta_{\text{mal}}) \in I$
11: $x_{\text{mal}} = x^* + \Delta_{\text{mal}}$
12: **return** $x_{\text{mal}}$

---

For CIFAR-10 the number of parameters in $x_{\text{mal}}$ is $3 \times 32 \times 32 = 3072$, but the largest number of shadow models used in the original LiRA paper is merely 256. Therefore, if we follow the loss Equation (3), $x_{\text{mal}}$ will overfit to shadow models and not be able to *generalize* to the target model.

To alleviate the computational burden and the overfitting problem, we make some modifications to the canary generation process. During optimization, we stochastically sample $b$ IN shadow models from $S_{\text{in}}$ and $b$ OUT shadow models from $S_{\text{out}}$ for each iteration, where $b < \min(n, m)$. This is equivalent to stochastic mini-batch training for batch size $b$, which might be able to help the query generalize better (Geiping et al., 2021). We find that such a mini-batching strategy *does* reduce the required computation, but it *does not* completely solve the overfitting problem. An attacker can easily find a $x_{\text{mal}}$ with a very low loss on Equation (3), and perfect separation of confidence scores from IN models and OUT models. However, querying with such a canary $x_{\text{mal}}$ results in random confidence for the holdout shadow models, which indicates that the canary is also not generalizable to the unseen target model.

To solve this, instead of searching for $x_{\text{mal}}$ on the whole feasible data domain, we initialize the adversarial query with the target image or the target image with a small noise. Meanwhile, we

Table 1: **Main Results on Different Datasets.** For three datasets, `Canary` attacks are effective in both online and offline scenarios.

| | Online | | | | | |
|---|---|---|---|---|---|---|
| | CIFAR-10 | | CIFAR-100 | | MNIST | |
| | AUC | TPR@1%FPR | AUC | TPR@1%FPR | AUC | TPR@1%FPR |
| LiRA | 74.36 | 17.84 | 94.70 | 53.92 | 56.28 | 3.95 |
| Canary | 76.25 | 21.98 | 94.89 | 56.83 | 58.12 | 5.23 |
| $\Delta$ | +1.89 | +4.14 | +0.19 | +2.91 | +1.84 | +1.28 |
| | Offline | | | | | |
| | AUC | TPR@1%FPR | AUC | TPR@1%FPR | AUC | TPR@1%FPR |
| LiRA | 55.40 | 9.85 | 79.59 | 42.02 | 50.82 | 2.66 |
| Canary | 61.54 | 12.60 | 82.59 | 44.78 | 54.61 | 3.06 |
| $\Delta$ | +6.14 | +2.75 | +3.00 | +2.76 | +3.79 | +0.40 |

add an $\varepsilon$ bound to the perturbation between $x_{\mathrm{mal}}$ and $x^*$. Intuitively, the hope is that $x_{\mathrm{mal}}$ and $x^*$ now share the same loss basin, which prevents $x_{\mathrm{mal}}$ from falling into a random, suboptimal local minimum of Equation (3). We summarize our complete algorithm `Canary` in Algorithm 1. In the offline case, we remove line 8 and only use OUT models during the optimization. We also illustrate our reasoning in Figure 1, where we visualize how the adversarially trained queries might alter the original queries' decision boundaries and provide more confident and diverse predictions.

Once a suitable canary has been generated, we follow the same metric-based evaluation strategy described in Section 3.2 but replace $(x^*, y^*)$ with $(x_{\mathrm{mal}}, y^*)$.

## 4 EXPERIMENTS

In this section, we first show that the `Canary` attack can reliably improve LiRA results under different datasets and different models for both online and offline settings. Further, we investigate the algorithm thoroughly through a series of ablation studies.

### 4.1 EXPERIMENTAL SETTING

We follow the setting of Carlini et al. (2022a) for our main experiment on CIFAR-10 and CIFAR-100 for full comparability. We first train 65 wide ResNets (WRN28-10) (Zagoruyko & Komodakis, 2016) with random even splits of 50000 images to reach 92% and 71% test accuracy for CIFAR-10 and CIFAR-100 respectively. For MNIST, we train 65 8-layer ResNets (He et al., 2016) with random even splits to reach 97% test accuracy. During the experiments, we report the average metrics over 5 runs with different random seeds. For each run, we randomly select a model as the target model and remaining 64 models as shadow models, and test on 5000 random samples with 10 queries.

For the hyperparameters in the `Canary` attack, we empirically choose $\varepsilon = 2$ for CIFAR-10 & CIFAR-100 and $\varepsilon = 6$ for MNIST, which we will ablate in Section 4.4. We sample $b = 2$ shadow models for each iteration and optimize each query for 40 optimization steps using Adam (Kingma & Ba, 2014) with a learning rate of 0.05. For $\mathcal{L}$ and $\mathcal{L}_{\mathrm{out}}$, we choose to directly minimize/maximize the logits before a softmax on the target label. All experiments in this paper are conducted by one NVIDIA RTX A4000 with 16GB of GPU memory, which allows us to load all shadow models and optimize 10 adversarial queries at the same time, but the experiments could be done with a smaller GPU by optimizing one query at a time or reloading the subsample of models for each iteration.

### 4.2 EVALUATION METRICS

In this paper, we mainly report two metrics: AUC (area under the curve) score of the ROC (receiver operating characteristic) curve and TPR@1%FPR (true positive rate when false positive rate is 1%). One can construct the full ROC by shifting the probability threshold of the attack to show the TPR under each FPR. The AUC measures the average power of the attack. As mentioned in 2 an attacker might be more interested in TPR with low FPR, so we also specifically report TPR@1%FPR.

Table 2: **Results on Different Models Architecture.** `Canary` attacks are able to consistently outperform LiRA over different models. T@1%F stands for TPR@1%FPR.

| | **Online** | | | | | | | |
| --- | --- | --- | --- | --- | --- | --- | --- | --- |
| | **WRN28-10** | | **ResNet-18** | | **VGG** | | **ConvMixer** | |
| | AUC | T@1%F | AUC | T@1%F | AUC | T@1%F | AUC | T@1%F |
| LiRA | 74.36 | 17.84 | 76.29 | 17.05 | 75.94 | 20.48 | 75.97 | 16.58 |
| Canary | 76.25 | 21.98 | 76.93 | 19.34 | 77.63 | 20.87 | 76.39 | 17.05 |
| Δ | +1.89 | +4.14 | +0.64 | +2.29 | +1.69 | +0.39 | +0.42 | +0.47 |
| | **Offline** | | | | | | | |
| | AUC | T@1%F | AUC | T@1%F | AUC | T@1%F | AUC | T@1%F |
| LiRA | 55.40 | 9.85 | 55.15 | 6.97 | 49.96 | 9.77 | 54.42 | 7.96 |
| Canary | 61.54 | 12.60 | 64.09 | 11.58 | 65.55 | 15.16 | 62.22 | 9.93 |
| Δ | +6.14 | +2.75 | +8.94 | +4.61 | +15.59 | +5.39 | +7.80 | +1.97 |

## 4.3 CANARY ATTACKS HELP MEMBERSHIP INFERENCE

We show our main results in Table 1 for three datasets. `Canary` attacks are effective in both online and offline scenarios. The improvement of TPR@1%FPR is significant for all datasets. The difference is especially notable for online CIFAR-10, where we achieve a $4.14\%$ boost over the baseline LiRA (a relative improvement in TPR of $23\%$). In the case of online CIFAR-100, where the baseline already achieves a very high AUC, `Canary` attacks only provide an extra $0.19\%$ over the baseline. On average, `Canary` attacks are most powerful in the more realistic offline scenario. We gain over $3\%$ boost on AUC scores on all datasets and over $2.75\%$ TPR@1%FPR boost for CIFAR-10 and CIFAR-100.

Overall, the improvement on MNIST is relatively small. We believe this can be attributed to the lack of diversity for MNIST, which is known to make membership inference more challenging. In this setting, the difference between the decision boundaries of IN models and OUT models is less pronounced, so it is more difficult to make diverse and reliable queries. Despite these challenges, we still see improvement over LiRA in the offline case – the AUC score is close to random ($50.82\%$) for LiRA here and `Canary` attacks can improve this to $54.61\%$.

In addition to WRN28-10, we further verify the ability of `Canary` attacks for three other models architectures in CIFAR-10: ResNet-18 (He et al., 2016), VGG-16 (Simonyan & Zisserman, 2014), and ConvMixer (Trockman & Kolter, 2022). In Table 2, `Canary` attacks are able to consistently provide enhancement over different models. The performance of `Canary` attacks should be related to the reproducibility of the model architecture. If the model decision boundary is highly reproducible, the shadow models should share similar decision boundaries with the target model, and the adversarial query trained on the shadow models will be more transferable to the target model. Indeed, we see from Table 2 that models with higher reproducibility do correlate with more improvement for the online scenario, especially between WRN28-10 and ConvMixer, where WRN28-10 has higher reproducibility than ConvMixer (Somepalli et al., 2022). We include further experiments regarding reproducibility and `Canary` in Appendix A.1.

## 4.4 ABLATION EXPERIMENTS

In this section, we provide ablation experiments on several crucial hyperparameters of the discussed `Canary` attacks.

**Number of shadow models.** As described before, the number of shadow models is comparable to the number of data points in traditional machine learning. We test `Canary` attacks with 5 different numbers of shadow models: $4, 8, 16, 32$, and $64$. We see from Figure 2(a), that using more shadow models yields a higher true positive rate when the false positive rate is low. Interestingly, as the number of shadow models initially decreases, the overall performance drops slightly, but such an effect diminishes after the number of shadow models is greater than $2^4$.

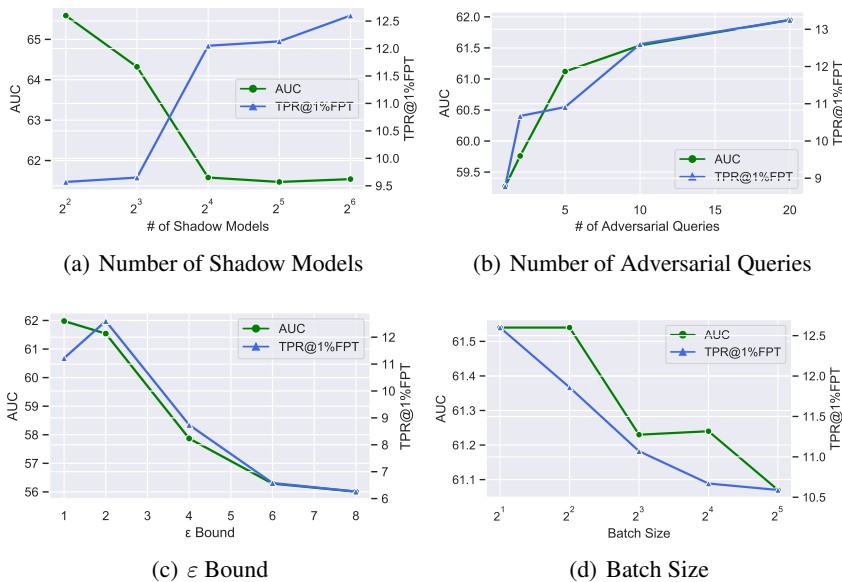

Figure 2: **Hyperparameter Ablation Experiments.** We provide ablation experiments on several crucial hyperparameters: number of shadow models, number of adversarial queries, $\varepsilon$ bound, and batch size.

**Number of queries.** Because of the stochasticity of optimization, different queries can fall into different minima of Equation (3), returning different sets of confidence scores and thus more ways to probe the target model. Therefore, it is essential to investigate how the number of queries affects the membership inference results. We plot the results in Figure 2(b). The ensemble of more adversarial queries consistently enhances both metrics, which means different queries indeed give different signals about the target model.

$\varepsilon$ **bound.** The choice of $\varepsilon$ is important, which is highly related to the transferability. As shown in Figure 2(c), the performance of Canary drops very fast after $\varepsilon = 2$. When $\varepsilon = 1$ the TPR@1%FPT is slightly lower than when $\varepsilon = 2$, which indicates that the perturbation within $\varepsilon = 1$ might be too small to be effective.

**Batch size.** In Figure 2(d), we test Canary with different batch sizes. Mini-batch strategy does improve the performance of Canary attacks. Especially for TPR@1%FPT, the difference is around 2% between the batch size of $2^1$ and $2^5$. Optimizing with a smaller batch size prevents the adversarial query from overfitting to the shadow models. Meanwhile, it massively reduces the GPU memory required for the gradient graph, which is a win-win situation for the attacker.

**Choice of Objectives for $L$ and $L_{\text{out}}$.** The choice the target objectives $L$ and $L_{\text{out}}$ is also crucial to the generalization of Canary attacks. We test six different objectives to create adversarial queries: 1) CE/reverse CE. 2) CE/CE on a random label other than the true label. 3) CW (Carlini & Wagner, 2017)/reverse CW. 4) CW/CW on a random label. 5) Directly minimize the scaled log score/maximize the scaled log score. 6) Directly minimize the pre-softmax logits of the true label/maximize the pre-softmax logits of the true label. We show the results in Table 3.

During the experiment, for all objectives above, we can easily get very low losses at the end of the optimization, and create Canary queries that perfectly separate the training shadow models. Surprisingly, minimizing/maximizing the pre-softmax logits gives us the biggest improvement, even though it does not explicitly suppress the logits for other labels like other objectives do. Overall, any other choices can also improve the baseline in the online scenario. However, in the offline scenario, only CW/CW and pre-softmax logits provide improvements to TPR@1%FPR.

## 4.5 DIFFERENTIAL PRIVACY

We now challenge Canary attacks with differential privacy (Abadi et al., 2016). Differential privacy is designed to prevent the leak of information about the training data. We evaluate Canary attacks in two settings. The first setting only uses norm bounding, where the norm bounding $C = 5$

Table 3: **Results with Different Objectives.** We evaluate `Canary` attacks on different objectives. Directly minimizing/maximizing the pre-softmax logits gives the biggest improvement in both the online and offline settings.

|  | Online | | Offline | |
|---|---|---|---|---|
|  | AUC | TPR@1%FPR | AUC | TPR@1%FPR |
| LiRA | 74.36 | 17.84 | 55.40 | 9.85 |
| CE/r. CE | 75.55 | 19.85 | 56.83 | 9.22 |
| CE/CE | 75.55 | 19.89 | 59.23 | 9.77 |
| CW/r. CW | 75.37 | 19.97 | 56.57 | 9.26 |
| CW/CW | 75.67 | 20.99 | 59.27 | 11.30 |
| Log. Logits | 75.82 | 20.01 | 59.16 | 8.04 |
| Logits | **76.25** | **21.98** | **61.54** | **12.60** |

Table 4: **Results under Differential Privacy.** In both cases, the norm clipping is 5. Even when the target model is trained with differential privacy, `Canary` attacks reliably increase the success of membership inference.

|  |  | Online | | Offline | |
|---|---|---|---|---|---|
|  |  | AUC | TPR@1%FPR | AUC | TPR@1%FPR |
| $\varepsilon = \infty$ | LiRA | 66.25 | 9.41 | 56.12 | 3.27 |
|  | Canary | 67.17 | 9.93 | 59.73 | 4.41 |
|  | $\Delta$ | +0.92 | +0.52 | +3.61 | +1.14 |
| $\varepsilon = 100$ | LiRA | 52.17 | 1.18 | 49.93 | 1.18 |
|  | Canary | 53.18 | 1.81 | 51.38 | 1.14 |
|  | $\Delta$ | +1.01 | +0.63 | +1.45 | -0.04 |

and $\varepsilon = \infty$, and in another setting, $C = 5$ and $\varepsilon = 100$. In order to follow the convention of practical differential privacy, we replace Batch Normalization with Group Normalization with $G = 16$ for ResNet-18.

We see in Table 4 that `Canary` attacks can provide some limited improvement. Both LiRA and `Canary` attacks are notably less effective when a small amount of noise $\varepsilon = 100$ is added during training, which is a very loose bound in practice. However, training with such a powerful defense makes the test accuracy of the target model decrease from $88\%$ to $44\%$. Differential privacy is still a very effective defense for membership inference attacks, but `Canary` attacks reliably increase the success chance of membership inference over LiRA.

## 5    CONCLUSION

We explore a novel way to enhance membership inference techniques by creating ensembles of adversarial queries. These adversarial queries are optimized to provide maximally different outcomes for the model trained with/without the target data sample. We also investigate and discuss strategies to make the queries trained on the shadow models transferable to the target model. Through a series of experiments, we show that `Canary` attacks reliably enhance both online and offline membership inference algorithms under three different datasets, four different models, and differential privacy.

Although `Canary` attacks perform very well in the above experiments, there are several relevant limitations. The optimization process for constructing the ensemble of canaries is markedly more computationally expensive than using data augmentations of the target data point as in Carlini et al. (2022b). Furthermore, effective optimization routines for queries could challenging, especially when considering future applications of this approach to discrete data, like text or tabular data. In principle, we believe it should be possible to devise a strategy to make adversarial queries transferable that do not require $\varepsilon$-bounds, but so far have found the method detailed in `Canary` to be the most successful approach.

ETHICS STATEMENT

Although our key goal is to develop a better membership inference algorithm to help protect data ownership, this technique might be used by a malicious party as a tool for breaching the privacy of the model trainer. On one hand, we find this acceptable, due to the inherent power imbalance between agents that train models and agents that own data. On the other hand, we believe that our results do not represent a fundamental shift in the capabilities of membership inference attacks.

ACKNOWLEDGEMENTS

This work was supported by the Office of Naval Research (#N000142112557), the AFOSR MURI program, DARPA GARD (HR00112020007), the National Science Foundation (IIS-2212182), and Capital One Bank.

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

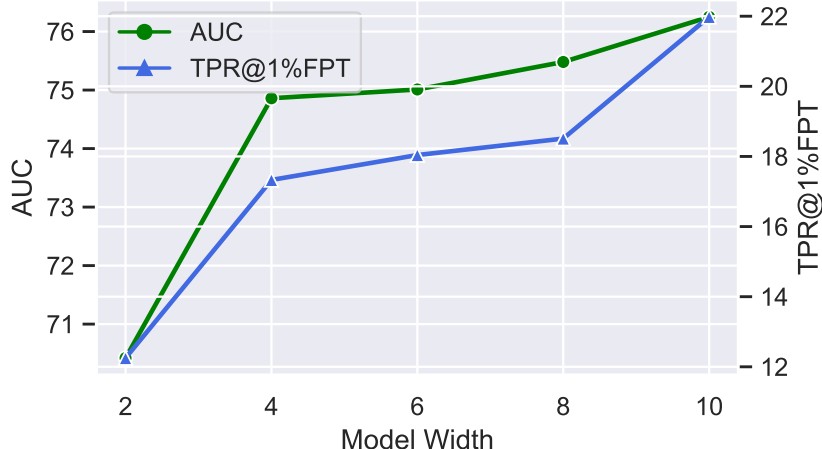

Figure 3: **Results with Different Widths of WRN28.**. We find that with increasing width of the WideResNet, attack success increases reliably. This is potentially related to an increase in repeatability of the decision boundaries of these models.

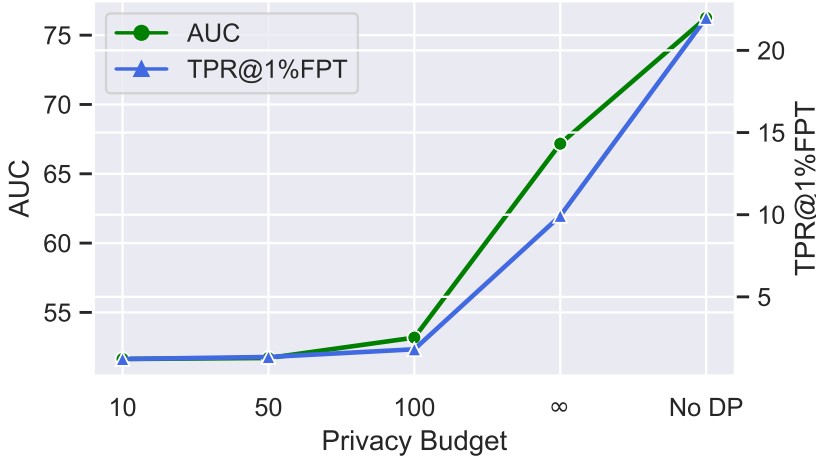

Figure 4: **More Results under Differential Privacy.** In all cases except No DP, the norm clipping is 5. Privacy Budget here refers to $(\varepsilon, 1e-5)$-DP, except for the last entry, which sets no budget and does not clip.

## A  APPENDIX

### A.1  ABLATION STUDY: MODEL WIDTH

Appendix A.1 shows the results on different model widths for WRN28. As claimed in Somepalli et al. (2022), wider networks have higher reproducibility. The performance of `Canary` correlates with the reproducibility (width) of the model.

### A.2  ABLATION STUDY: PRIVACY BUDGET

In Appendix A.2, we provide more results with more strict privacy budgets. When the $\varepsilon$ of differential privacy is 10, the AUC of the attack is very close to $50\%$ which is the random guess, and the TPR@1%FPR is almost zero. Differential privacy is still a very strong defense against membership inference.

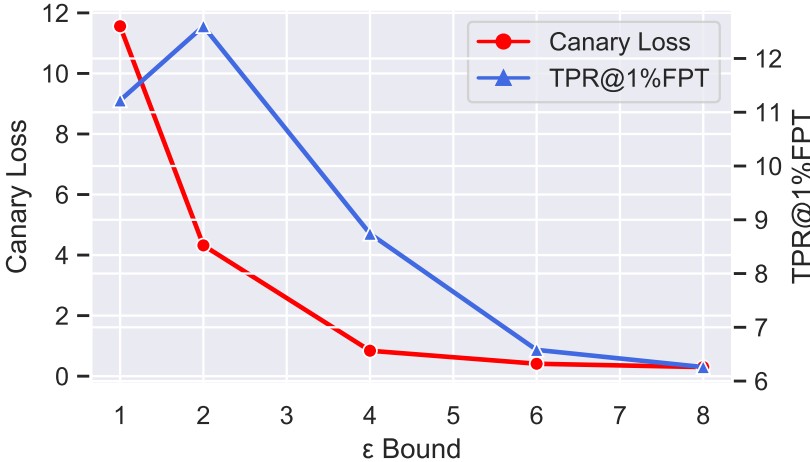

Figure 5: **Canary Loss** (i.e. the objective evaluated on the shadow models used to optimize the canary attack) plotted against **Performance** (evaluated on the unseen test model) for various values of $\varepsilon$. When $\varepsilon$ is too large, then the attack overfits and does not generalize to the unseen test model that is attacked during membership inference.

Table 5: **Comparison with Random Noise Perturbations.** A random noise perturbation in the same $\varepsilon$-ball as the canary does not increase membership success. In this sense, the optimized behavior of the canary attack is crucial.

|              | AUC   | TPR@1%FPR |
| ------------ | ----- | --------- |
| LiRA         | 74.36 | 17.84     |
| Random Noise | 74.30 | 17.87     |
| Canary       | 76.25 | 21.98     |

### A.3 CANARY LOSS V.S. PERFORMANCE

As shown in Appendix A.3, if we increase $\varepsilon$, the Canary objective monotonically decreases, evaluated on the "training" shadow models, but attack success peaks and then decreases.

### A.4 CANARY V.S. RANDOM NOISE

We test adding random noise within the same $\varepsilon$ ball on the original target image, as shown in Table 5. The attacker doesn't benefit from adding random perturbations.

### A.5 COMPUTATIONAL COST

Table 6 shows the average attack time in seconds on one single NVIDIA RTX A4000 over 5000 target images with a total of 64 shadow models.

Table 6: **Computational Cost in Seconds.** for generation of the attack and query into the model. Not included for both methods is the computational cost to create the array of shadow models.

|        | 1 Query | 2 Queries | 5 Queries | 10 Queries |
| ------ | ------- | --------- | --------- | ---------- |
| LiRA   | 0.26    | 0.27      | 0.36      | 0.63       |
| Canary | 1.03    | 1.04      | 1.44      | 2.44       |

