# OpenReview forum: "Canary in a Coalmine: Better Membership Inference with Ensembled Adversarial Queries"
_ICLR.cc/2023/Conference — ICLR 2023 notable top 25%_

### Official Review · Reviewer_7EZU · 2022-10-23

**Confidence:** 5
**Correctness:** 4
**Technical Novelty And Significance:** 3
**Empirical Novelty And Significance:** 3
**Recommendation:** 8

**Clarity, Quality, Novelty And Reproducibility:**

The overall quality of this paper is good. The idea is novel. The code is provided to reproduce the paper.

**Strength And Weaknesses:**

Strengths:
1. The idea of using adversarial attacks to optimize the MIA queries is quite novel.
2. Following the recent research in MIAs, the paper uses both AUC and FPR to measure the privacy threats.
3. The Canary attack outperforms the state-of-the-art LiRA attack.
4. The paper investigated practical attack scenarios in online and offline threat models. The effectiveness of the proposed Canary attack has been demonstrated in both threat models.
5. The paper investigates the impact of several critical hyperparameters in Canary attacks.
6. The paper is well-written.

Weaknesses:
1. The optimization process of the query generation introduces additional computational overhead. The attack performance is also related to the number of queries. It would be great to compare the computational cost with LiRA attack. In addition, how do we select perturbation bound ϵ in practice? Does the selection require additional cost?
2. The evaluation only considers differential privacy for privacy protection. The paper could discuss the performance of Canary on other state-of-the-art defenses.

**Summary Of The Paper:**

This paper presents a novel black-box membership inference attack against machine learning moles. The proposed Canary attack optimizes the queries in MIA using adversarial attacks and achieves good attack performance.

**Summary Of The Review:**

The paper presents a novel and effective membership inference attack. Using adversarial attacks in query optimization is novel. The proposed attack outperforms the state-of-the-art attack. The paper could include the computational cost and the experiments against other defenses in the paper.

---

> ### Author Response · Authors · 2022-11-15
> **Response**
>
> Thank you for your feedback and we appreciate your support for this submission. Below, we address specific questions that you brought up:
>
> > Computational cost
>
> The table below shows the average attack time in seconds on one single NVIDIA RTX A4000 over 5000 target images with a total of 64 shadow models. Due to the fact that in each iteration, we only sample a mini-batch of shadow models, the attack time is not extremely slow.
> |            | 1 Query | 2 Queries | 5 Queries | 10 Queries |
> |:----------:|:-------:|:---------:|:---------:|:----------:|
> |  **LiRA**  |  0.26   |   0.27    |   0.36    |    0.63    |
> | **Canary** |  1.03   |   1.04    |   1.44    |    2.44    |
>
> Ultimately, the main cost in both this work and LiRA is to generate all shadow models. The actual generation of the adversarial query pales in comparison to the computational effort of training, e.g. 64 models for both attacks.
>
> > In addition, how do we select perturbation bound ϵ in practice? Does the selection require additional cost?
>
> Thank you for bringing this up. We have added additional material to the Appendix in A.3 and Fig.5 to clarify the tradeoff between accuracy and overfitting that comes with the choice of $\varepsilon$. Briefly, when $\varepsilon$ is too small, then the adversarial query is not well optimized, but if $\varepsilon$ is too high, then the adversarial query overfits to the existing shadow models and does not generalize to unseen test models. The optimal $\varepsilon$ can be found via cross-validation, although in our work, we mostly focus on checking only $\varepsilon \in \{1,2\}$, for most experiments, which is a limited expense in computing (given that the existing shadow models can be re-used).
>
> |                 | $\varepsilon=$ 1 |   2   |  4   |  6   |  8   |
> |:---------------:|:-------------:|:-----:|:----:|:----:|:----:|
> |  **TPR@1%FPR**  |     11.22     | 12.60 | 8.74 | 6.58 | 6.26 |
> | **Canary Loss** |     11.56     | 4.32  | 0.84 | 0.41 | 0.3  |
>
> > Other state-of-the-art defenses
>
> We do think that strong differential privacy is widely regarded as the core defense against membership inference, so we have focused that section of the paper on DP in particular. Of course, this reflects our view of the community. What other defenses would you find especially interesting to compare to? We would be happy to include a discussion of the potential impact of other defenses in our work.
>
> Overall, we're glad you found this paper to contain a number of strengths and find the inclusion of additional measurements of computational costs helpful. Please let us know if you have additional questions or comments that we can address.

---

### Official Review · Reviewer_4F4t · 2022-10-26

**Confidence:** 3
**Clarity, Quality, Novelty And Reproducibility:** paper is clearly written and is easy …
**Correctness:** 3
**Technical Novelty And Significance:** 3
**Empirical Novelty And Significance:** 3
**Recommendation:** 8

**Strength And Weaknesses:**

Membership inference attacks are of great interest to the privacy and security community, and the paper proposes a method to improve LiRA by learning the query vectors. I find the proposed method interesting, and of potential interest to the community. Paper is clearly written and is easy to understand.

My only comment is that it would be interesting to see the metrics with more stringent privacy budget (for differential privacy), example: $\varepsilon \in [10,20,\cdots,100]$, it will also add another practical dimension to the paper, proving strong DP protections against membership inference.

**Summary Of The Paper:**

Paper proposes a membership inference method based on learning query vectors, and shows that it improves the current state-of-the-art methods.

**Summary Of The Review:**

I like the method proposed and view the paper having a positive impact.

---

> ### Author Response · Authors · 2022-11-15
> **Response**
>
> Thank you for your feedback and we appreciate your interest in this submission.
>
> > More stringent privacy budget
>
> We added more experiments with lower privacy budgets in Appendix A.2 and Fig.4. Fig.4 shows when the $\varepsilon$ of differential privacy is 10, the AUC of the attack is very close to 50% which is the random guess, and the TPR@1%FPT is almost zero. We can clearly see how effective differential privacy is in protecting training data privacy.
>
> We're glad that you liked the proposed method of improved membership inference with adversarial queries and would be happy to answer further questions and comments.

---

> > ### Comment · Reviewer_4F4t · 2022-11-15
> > **Re: Response**
> >
> > Thank you for the response and for adding the experiment. I think the label for x-axis for Figure 5 needs revision.

---

> > > ### Author Response · Authors · 2022-11-15
> > > **Thank you for the feedback**
> > >
> > > Thank you for pointing this out. We have updated the draft and improved the captions of Fig.3,4,5 to avoid confusion (but are unsure how to improve the x-axis labels in particular). Please let us know if you have further questions or suggestions!

---

> > > > ### Comment · Reviewer_4F4t · 2022-11-15
> > > > **Re: Response**
> > > >
> > > > Thank you. That was my mistake. I confused $\varepsilon$ with privacy budget.

---

### Official Review · Reviewer_3Ke9 · 2022-10-26

**Confidence:** 2
**Correctness:** 3
**Technical Novelty And Significance:** 3
**Empirical Novelty And Significance:** 3
**Recommendation:** 8

**Clarity, Quality, Novelty And Reproducibility:**

The presentation of this work is well organised and the main contribution is clearly conveyed. Besides, the paper provides enough details to reproduce the experimental results.

**Strength And Weaknesses:**

Strong point:

Perturbing the target data point to boost the membership inference precision is a novel and interesting idea. The core idea behind is to make use of the perturbed variant of the target data point to facilitate the attack. Even though the loss of the target data point may hardly vary if the target data point is considered to be an in- or out-class sample, the perturbed variants within the epsilon-neighbourhood of the target data point can potentially unveil more information to identify the membership.

Weakness:
1. In Table.2, the AUC and the TPR values (given the FPR level) of membership inference attack do not provide a monotonic decreasing as the reproductivity of the target model decreases (as claimed in the paper, the target models in Table.2 are given in a descending order of the model reproductivity.)

2. On page 8, why does the attack performance become worse if the bound epsilon becomes larger or smaller than 2? Any explanation to the observation?  Is there any potential trade-off related to the choice of epsilon ?

3. More theoretical study regarding the effectiveness of the adversarial query could further consolidate this work. The adversarial perturbation added to the target data point may help show the loss distribution within the epsilon-neighborhood of the target data point. Though the loss on the target data point may not be rich enough for a successful membership inference attack, the difference of the loss distribution may provide a better lens to differentiate two different membership classes. For the moment, the reasonability behind Eq.3 is still not very clear.

**Summary Of The Paper:**

In this work, an adversarial query method is proposed to boost the likelihood test-driven membership inference attack (LiRA). Instead of using the target data point as a query directly, the adversary first estimates a variant of the target data point by slightly perturbing the original target data point and minimising the fitting loss of the in-class hypothesis and out-class shadow models. The attacker then uses the estimated variant as an adversary query to infer the membership of the target data point. Experimental study shows that using the adversarial query can indeed improve the attack performances of LiRA in the low FPR regime over different datasets.

**Summary Of The Review:**

I like the idea of injecting additional perturbation to boost the data privacy leak-driven attack. It offers an alternative way to differentiate the membership class given a target data, other than compare the membership likelihood scores as in LiRA. However, the major bottleneck of this work is the lack of explanation and discussions regarding the experimental observations. As pointed out in the weakness, there is apparently a trade-off existing in the choice of epsilon. This is also related to the theoretical justification of the rationality of Eq.3. Without these discussions, it is difficult to understand why and when the proposed method can improve the inference performances.

---

> ### Author Response · Authors · 2022-11-15
> **Response**
>
> Thank you for your feedback. Below, we address specific points you raised.
>
> > Model reproductivity vs Canary performance
>
> Thank you for bringing this up, we've clarified the caption of Table 2 and removed this statement. We agree that the connection to the work cited therein may be a bit tenuous, given that all models investigated here have a range of different hyper-parameters and are harder to rank by the reproducibility of decision boundaries.
>
> Instead, to show that reproducibility matters in a more controlled experiment, we have added a series of new experiments in Appendix A.1, where we scale a ResNet in width. When a model width grows, the reproducibility of the model also increases. As shown in Fig.3 in the Appendix, the Canary attack has a strictly better performance as the model width increases.
>
> > Trade-off related to the choice of epsilon
>
> In terms of $\varepsilon$ choices, we actually find a very natural bias-variance trade-off. For $\varepsilon=1$, the perturbation is too small, and the information that the Canary gains over the baseline is limited. However, if $\varepsilon$ is too large, (e.g. $\varepsilon=8$), then the Canary query images face an overfitting problem. As the number of shadow models is small (e.g. 64) compared to the number of free parameters in the canary (3072 pixels), the Canary images overfit to the shadow models but do not generalize to the unseen test model. As shown in the table below and append A.3 and Fig.5, if we increase $\varepsilon$, the Canary objective monotonically decreases, evaluated on the "training" shadow models, but attack success peaks and then decreases. Therefore, $\varepsilon$ is an important hyperparameter for these Canary attacks. The attacker can optimize $\varepsilon$ using a validation set of held-out shadow models, for example via cross-validation.
>
> |                 | $\varepsilon=$ 1 |   2   |  4   |  6   |  8   |
> |:---------------:|:-------------:|:-----:|:----:|:----:|:----:|
> |  **TPR@1%FPR**  |     11.22     | 12.60 | 8.74 | 6.58 | 6.26 |
> | **Canary Loss** |     11.56     | 4.32  | 0.84 | 0.41 | 0.3  |
>
> > The reasonability behind Eq.3
>
> We agree with your general hypothesis here. We believe that via optimization of the adversarial query, we can find a position in the model output space than can better differentiate in-group and out-group models. We think this idea is intuitively appealing when considering "average-case" data points (like a "very average cat image"), that are unlikely to lead to different predictions in in-group or out-group by themselves but might be found by investigating other locations in the model output space.
>
> We've added additional results to the Appendix concerning your question about samples from the epsilon-neighborhood of the target. We test adding random noise within the same $\varepsilon$ ball on the original target image. The table below shows that the attacker doesn't benefit from adding random perturbations, i.e. random samples from the epsilon ball, and this hence reflects that the perturbation generated by Eq.3 is meaningful, and represents adversarial behavior.
> |                  |  AUC  | TPR@1%FPR |
> |:----------------:|:-----:|:---------:|
> |     **LiRA**     | 74.36 |   17.84   |
> | **Random Noise** | 74.30 |   17.87   |
> |    **Canary**    | 76.25 |   21.98   |
>
>
> Thank you for your interest in this submission. We hope that the additional experiments above elucidate the questions of the trade-off in the choice of epsilon and justify the choice of Eq.(3). Please let us know if you have other questions and comments that we can address.

---

> > ### Comment · Reviewer_3Ke9 · 2022-11-15
> > **Thanks for the response.**
> >
> > Thanks for the detailed explanations. Though I am keen to more theoretical justification of the efffectiveness of this work (may be related to the out-of-distribution generalization capability of a classifier? ), I think the paper is really solid in the current form. Therefore I will raise my rating score accordingly.

---

### Decision · Program_Chairs · 2023-01-20

**Decision:**

Accept: notable-top-25%

**Justification For Why Not Higher Score:**

While all the reviewers felt that the experiments were adequate, nobody found the work to be a breakthrough or remarkably interesting. One reviewer wished for better theoretical motivation.

**Justification For Why Not Lower Score:**

N/A

**Metareview: Summary, Strengths And Weaknesses:**

This is a work about a black-box test to determine whether a certain example was in the training set. A novel "Canary" algorithm is developed  which uses adversarial queries. It is experimentally compared with baselines from prior work, specifically LiRA.

The reviewers found the idea of perturbing the target data point to boost the membership inference precision to be interesting and creative. The experiments were adequate and conveyed the main points of the result. One reviewer called for further theoretical study regarding the effectiveness of the adversarial queries.

**Note From Pc:**

if the above contains the word "oral" or "spotlight" please see: "oral" presentation means -> notable-top-5% and "spotlight" means -> notable-top-25%. As stated in our emails, we are disassociating presentation type from AC recommendations